# Transcription Factors in the Fungus *Aspergillus nidulans*: Markers of Genetic Innovation, Network Rewiring and Conflict between Genomics and Transcriptomics

**DOI:** 10.3390/jof7080600

**Published:** 2021-07-25

**Authors:** Oier Etxebeste

**Affiliations:** Laboratory of Biology, Department of Applied Chemistry, Faculty of Chemistry, University of the Basque Country (UPV/EHU), 20018 San Sebastian, Spain; oier.echeveste@ehu.eus; Tel.: +34-943-018517

**Keywords:** transcription factor, transcriptional regulation, transcriptomics, gene duplication, gene regulatory network, network rewiring, filamentous fungi, *Aspergillus nidulans*

## Abstract

Gene regulatory networks (GRNs) are shaped by the democratic/hierarchical relationships among transcription factors (TFs) and associated proteins, together with the cis-regulatory sequences (CRSs) bound by these TFs at target promoters. GRNs control all cellular processes, including metabolism, stress response, growth and development. Due to the ability to modify morphogenetic and developmental patterns, there is the consensus view that the reorganization of GRNs is a driving force of species evolution and differentiation. GRNs are rewired through events including the duplication of TF-coding genes, their divergent sequence evolution and the gain/loss/modification of CRSs. Fungi (mainly Saccharomycotina) have served as a reference kingdom for the study of GRN evolution. Here, I studied the genes predicted to encode TFs in the fungus *Aspergillus nidulans* (Pezizomycotina). The analysis of the expansion of different families of TFs suggests that the duplication of TFs impacts the species level, and that the expansion in Zn2Cys6 TFs is mainly due to dispersed duplication events. Comparison of genomic annotation and transcriptomic data suggest that a significant percentage of genes should be re-annotated, while many others remain silent. Finally, a new regulator of growth and development is identified and characterized. Overall, this study establishes a novel theoretical framework in synthetic biology, as the overexpression of silent TF forms would provide additional tools to assess how GRNs are rewired.

## 1. Introduction

Transcription factors (hereafter referred to as TFs) are proteins that are capable of binding DNA at cis-regulatory sequences (CRSs) of target genes. TF-CRS binding repress or promote transcription by RNA polymerase. The biological impact of most TFs cannot be completely assessed individually but collectively, as interacting units of a network or interconnected networks of TFs, associated proteins and CRSs, which are generally known as gene regulatory networks or GRNs [1]. The architecture of GRNs ranges from simple, such as that controlling mating-type in yeasts [2], to complex, multistep hierarchical and democratic relationships formed by multiple TFs and associated proteins, such as the biofilm network of *Candida albicans* or the embryonic stem cell network of *Mus musculus* (compared by [3]).

GRNs control all cellular processes, how cells interact with the environment and how they adapt to change. Fundamentally, they determine the growth and developmental patterns of all species. Considering that every single cellular, organ, tissue or organismal morphology is the phenotypic manifestation of the specific architectures of and connections among GRNs, the introduction or removal of regulatory units within GRNs, or modifications in the connections among regulatory units, have the potential to render modified developmental phenotypes. Thus, it is generally accepted that the rewiring of GRNs is an important driving force of species evolution and differentiation [1].

There are different mechanisms for modifying the structure of GRNs [4]. The emergence of new regulators (rather than the incorporation of individual target genes: “regulator-first” model, see references within [3]) is one of the major events, and occurs mainly through the duplication of genes coding for TFs and the divergent evolution of each paralog sequence. Amino acid changes in TFs can precede gains in CRSs [5], leading to neofunctionalization (the development of new functions) or subfunctionalization (retention of a subset of the functions of the original paralog) [3,4].

The kingdom fungi is composed of 2–4 million species [6]. The most primitive clades of the fungal tree of life are those of Blastocladiomycota, Chytridiomycota and Zoopagomycota, while the three main clades correspond to Mucoromycota, Ascomycota and Basidiomycota [7]. While species belonging to the class of Saccharomycetes reproduce mainly asexually by budding, most species of these three clades form filamentous multicellular colonies and more complex reproductive structures. Saccharomycetes have traditionally been used to study how GRNs are shaped and evolve [2,3,5]. Recent comparative studies carried out in Basidiomycota and Ascomycota have shown that TFs controlling (asexual and/or sexual) developmental patterns emerged gradually in evolution [8,9,10]. Furthermore, the formation of main clades, such as that of Basidiomycota or that of the subphylum Pezizomycotina, was accompanied by a boost in the number of TF-coding genes. This stepwise emergence of TFs has remarkable implications regarding, for example, the control of developmental patterns. In the order Eurotiales, the transcription factor BrlA plays a role as a bottleneck in the formation of asexual structures known as conidiophores [11]. BrlA emerged much later than other upstream and downstream TFs of the same GRN [12,13], suggesting that it caused rewiring events in pre-existent ancient networks [9].

The starting point of this work is an update of the inventory of potential TF-coding genes elaborated by Wortmann and colleagues in the genome of the filamentous reference fungus *Aspergillus nidulans* [14]. Comparative searches show that there are TFs with a within-genus conservation pattern, suggesting that the stepwise emergence of TFs mentioned above impacts the species level. Multiple TFs, such as the C2H2-type BrlA, SteA and MsnA, possibly share a common ancestor, showing that duplication events are recurrent. This seems to be a general trend in fungi, and is a predominant mechanism in the most represented family of DNA-binding domains (DBDs) of Ascomycetes: Zn2Cys6 clusters and fungal-specific TF domains. The analysis of several RNA-seq experiments carried out using *A. nidulans* as the reference system and their comparison with gene models in the FungiDB database suggest that the annotation of a high proportion of genes encoding TFs should be updated, and that many others tend to remain silent. A new regulator of *A. nidulans* growth and sexual/asexual developmental cycles, AN2001, is also identified and the phenotype of the corresponding null mutant characterized. Overall, results suggest that filamentous fungal genomes constitute valuable resources to study how the set of regulatory proteins and the regulatory networks they form are organized and evolve. Furthermore, modification of the coding sequences of primarily silent TF forms, their overexpression and their functional assessment, would provide additional tools for studying mechanisms of GRN rewiring.

## 2. Materials and Methods

### 2.1. Database of the Potential TFs Encoded by the A. nidulans Genome

A preliminary list of *A. nidulans* genes coding for TFs was obtained from Wortman and colleagues [14], and was updated with the addition of genes annotated as coding for TFs in the FungiDB database (releases 47–50) [15] but originally not included in the reference list. The “*protein features and properties: InterPro domain*” search function was used with this aim in the FungiDB database. The lists provided by the FungiDB database were compared to that previously published. The sequence and predicted functional domains of those potential TFs not included in the list by Wortman et al. were checked again in the FungiDB database and with the bioinformatics tools described in Section 2.2, in order to include or exclude them in the present work. Amino acid sequences of each potential TF were retrieved from the FungiDB database. Genes annotated as pseudogenes and containing stop signals in their predicted coding sequence were not considered.

### 2.2. Bioinformatic Analyses

Annotated gene and protein sequence analyses were carried out in FungiDB and AspGD (*Aspergillus* genome) databases. The presence of putative signal peptides was analyzed using Interpro [16] and SignalP 5.0 [17]. Presence of putative transmembrane (TM) domains within protein sequences was analyzed using Interpro, TopCons2 [18], TMHMM v2.0 [19] and Phobius [20]. Default settings were used in all these websites. Protein BLAST (basic local alignment search tool) searches were carried out at the NCBI’s (National Center for Biotechnology Information) website (https://blast.ncbi.nlm.nih.gov/Blast.cgi?PAGE=Proteins, accessed on 1 December 2020). Protein sequences were aligned using Clustal Omega [21]. Phylogenetic trees were generated with MEGAX, using neighbor-joining or maximum-likelihood methods and Felsenstein’s bootstrap test of phylogeny, which is evaluated using Efron’s resampling technique [22]. Phylogenetic trees (MEGAX) were edited using iTOL version 6 [23]. Homology searches were carried out in the HMMER web server (versions 3.3.1 and 3.3.2), which bases biosequence analyses on Hidden Markov Models [24]. Full fasta sequences of the hits identified by HMMER were retrieved and analyzed in the NCBI BLAST site (align two or more sequences) to obtain query coverage values. CD-Hit (http://weizhong-lab.ucsd.edu/cdhit_suite/cgi-bin/index.cgi?cmd=Server%20home, accessed on 1 December 2020) was used (sequence identity cut-off of 0.3) to identify and cluster possible paralogs among Zn2Cys6 and C2H2-type TFs [25].

### 2.3. RNA-Seq Data Analyses

Expression values of each gene annotated as coding for TFs were obtained as FPKM (fragment per kilobase of transcript per million mapped reads) or TPM (transcripts per kilobase million) values from RNA-seq experiments available in the literature and corresponding to different genetic backgrounds and culture conditions (see Data Availability Statement). FPKM values corresponding to (1) the use of different nitrogen sources or nitrogen starvation (quantitative data in this case correspond to a single sample per condition and do not include biological replicates; see comments in the last section of results), (2) a null *veA* genetic background, (3) a null *mtfA* background or (4) a null *clrB* genetic background, were retrieved from the FungiDB database [26,27,28]. TPM values corresponding to (5) samples comparing alkaline pH (8.0) or sodium stress (1M NaCl) to standard culture conditions (*Aspergillus* minimal medium or AMM), in both the wild-type and a null *sltA* background, were obtained from [29]. FPKM values comparing (6) samples collected before and after the induction of conidiation (vegetative growth, VG and asexual development, AD), both in a null *flbB* strain and its isogenic wild-type reference were obtained from [30,31]. Raw RNA-seq data deposited under accession numbers (7) E-MTAB-6996 (null *sclB* background) [32], (8) PRJNA588808 (null *vosA* background) [33], (9) GSE72316 (null *osaA* background) [34] and (10) PRJNA294437 (use of waste steam-exploded sugarcane bagasse as a nutrient source) [35], were retrieved and processed at the Galaxy website [36]. *bam* files were generated from *fastq* files, the reference genome file *A_nidulans_FGSC_A4_version_s10-m02-r03_chromosomes.fasta*, and the gene model *gtf* file *A_nidulans_FGSC_A4_version_s10-m02-r03_features_with_chromosome_sequences.gtf* (both downloaded from the AspGD database) using RNA Star (2.7.5b) [37]. Stringtie (version 2.1.1) [38] or CuffNorm (Galaxy version 2.2.1.3) [39] were used to generate FPKM-based or TPM-based expression tables, using the previously generated *bam* files and default job resource parameters.

To check the annotation of genes potentially coding for TFs and the presence or absence of predicted introns, *bam* files corresponding to RNA-seq data published by us [29,30,31] were analyzed using Integrative Genome Viewer (IGV version 2.7.1 [40]), in combination with the set of transcriptomics data available in the FungiDB database. GO (gene ontology) enrichment analysis were carried out using ShinnyGO [41]. Heatmaps were generated using Heatmapper [42] and Morpheus, a versatile matrix visualization and analysis software developed by the Broad Institute (https://software.broadinstitute.org/morpheus accessed on 1 December 2020).

### 2.4. Generation of Recombinant Strains of A. nidulans

The fusion PCR procedure [43] was used to generate linear DNA constructs, in which the region coding for the C2H2-type DBD of BrlA (codons for F320-H375) had been replaced by the region coding for the C2H2 DBD of MsnA (codons for T463-H515). The rest of the *brlA* coding sequence (*brlA* has no introns) was maintained as in the wild-type version (codons for M1-Q319 and S376-E432) and the beginning of *brlAα* and *brlAβ* transcripts [44] were maintained as in the wild type. First, a *brlA::ha_3x_::pyrG::brlA_3’-UTR_* construct was generated. With that aim, 3 fragments were amplified and fused: (1) the promoter and the coding region of *brlA* (3.6 Kb; oligonucleotides BrlA-PP1 and BrlA-GSP2) (Appendix A); (2) the sequence coding for the HA_3x_ tag plus *pyrG^Afum^* as selection marker (2.2 Kb; oligonucleotides BrlA-GFP1 and BrlA-GFP2); and (3) 1.5 Kb of the 3′-UTR (untranslated) region of *brlA* (1.6 Kb; oligonucleotides BrlA-GSP3 and BrlA-GSP4).

This construct was used as a template in new PCR reactions to amplify and fuse: (1) the promoter of *brlA* and its coding sequence until the codon for Q319 (3.4 Kb; oligonucleotides BrlA-PP1 and BrlA-ZnFDw; Appendix A); (2) the region coding for the C2H2 DBD of MsnA (0.2 Kb; genomic DNA of a wild-type strain as a template, oligonucleotides MsnA_ZnF_forBrlAUp and MsnA_ZnF_forBrlADw); and (3) the end of the coding sequence of *brlA* (codons for S376-E432) plus *ha_3x_::pyrG^Afum^* and the 3′-UTR region of *brlA* (3.8 Kb; oligonucleotides BrlA-ZnFUp and BrlA-GSP4).

Fusion PCR cassettes, *brlA::ha_3x_::pyrG^Afum^::brlA_3′-UTR_*, used as a control to confirm that the HA_3x_ tag did not impede BrlA function, and *brlA_M1-Q319_::msnA_T463-H515_::brlA_S376-E432_::ha_3x_::pyrG^Afum^*::*brlA_3′-UTR_*, were used to transform protoplasts of the reference strain TN02A3 (*pyrG89; argB2; pyroA4,* Δ*nkuA::argB; veA1*; [45]), following the procedures previously described by [46,47]. BrlA::HA_3x_ transformants showed a wild-type conidiating phenotype, while BrlA-MsnA chimera transformants showed an aconidial phenotype. Genomic DNA of selected transformants was extracted as previously described by us [48], and the maintenance of the reading frame in the *brlA-msnA* chimera was confirmed by Sanger sequencing.

The fusion PCR procedure was used to generate synthetic, linear DNA cassettes for the generation of null mutants of *AN10192* (strain BD1421), *AN1500* (strain BD1423), *AN2001* (strain BD1425), *AN4324* (strain BD1427) and *AN4527* (strain BD1430). Three fragments were amplified and fused in each case: (1) 1.5 Kb of the 5′-UTR region of the targeted gene, using genomic DNA as a template and oligonucleotides PP1 and PP2 (Appendix A); (2) *pyrG^Afum^* as the selection marker (oligonucleotides SMP1 and GFP2, and a plasmid bearing the selection marker as template); and (3) the 3′-UTR region of the targeted gene, using oligonucleotides GSP3 and GSP4 and genomic DNA as the template. Each cassette was used to transform protoplasts of strain TN02A3. After extraction of DNA of candidate transformants, diagnostic PCRs using oligonucleotide pairs sPP1/sGSP4, sPP1/GFP2 or SMP1/sGSP4 were carried out to confirm homologous recombination at the targeted *loci*.

## 3. Results

### 3.1. The Set of Potential TFs in the Ascomycete Fungus Aspergillus nidulans Is Dominated by Binuclear Zinc Clusters

*Aspergillus nidulans* is an ascomycete of the Eurotiomycetes class (subphylum Pezizomycotina). It has been traditionally used as a reference system for the study of filamentous fungal biology [49]. Thus, a large number of TFs have been characterized functionally and their genetic/molecular inter-relationships have been determined. TFs and GRNs controlling in fungi, e.g., stress-response, carbon and nitrogen metabolism, light sensing and the induction/progression of developmental cycles were characterized for the first time in this Aspergillus species. Taking advantage of this knowledge, recent works have focused on the study of how some of these GRNs were structured during the evolution of the kingdom fungi [9,12,13]. An interesting feature of the TFomes of Ascomycota is the remarkable expansion of zinc cluster-type TFs, while in Basidiomycota and early diverging fungi, C2H2-type TFs are the most represented family [50,51]. To explore new features of its TFome, the conservation and expression patterns of the genes encoding these TFs, and uncover new traits of the organization and evolution of GRNs in the kingdom fungi, firstly, I updated the list of TFs potentially encoded in the genome of *A. nidulans*. With this aim, the list published by Wortman and colleagues was used as a reference [14] and updated (Appendix A) based on the lists of TFs given by the FungiDB database [15]. The information available about each predicted TF, the extension of their regulatory domains (IPD: Interpro Regulatory Domains) and their amino acid sequences were retrieved from the FungiDB database and the InterPro server [16]. Appendix A lists the 923 IPDs elaborated in the predicted 520 TFs, together with their full length and IPD amino acid sequences. The tables in Figure 1A show the number of TFs and IPDs in each family. The updated distribution shows that *A. nidulans* follows the same trend as the Ascomycota species [50,51]. Zn2Cys6 family TFs are 63.7% (331 of 520) of the predicted TFome, and Zn2Cys6 IPDs are 36.0% (332 of 923) of the total number of predicted DNA-binding domains. Both are followed by Fungal-specific regulatory TFs/IPDs, most of the times associated with zinc clusters, and C2H2-type TFs/IPDs (14.0%, 73 of 520, and 19.7%, 182 of 923, of C2H2-type TFs and IPDs, respectively) (Figure 1A). The values given for Zn2Cys6-type TFs and IPDs were obtained after the observation that the annotation of specific genes (e.g., *AN11962*, *AN1705*, *AN5464*, *AN5510*, *AN7076*, *AN7584*, *AN8103* or *AN8501*) was probably incorrect, as their 5′-UTR or 5′-coding regions included non-predicted or non-processed introns, as well as regions coding for Zn2Cys6 IPDs (see next section). This suggests that the percentage of Zn2Cy6-type TFs or IPDs coded by the genome of *A. nidulans* is even higher than initially predicted.

Of a total of 520 potential TFs, 118 (22.7%) had a standard name, meaning that they have been functionally characterized to some extent (Appendix A; Figure 1B). This suggests that 77.3% (402 out of 520) of the predicted TFome of *A. nidulans* remains to be functionally characterized. The FungiDB database predicted the presence of a signal peptide in 16 potential TFs (3.1%) while SignalP was much more restrictive (only 4; 0.8%) (Appendix A; Figure 1B), suggesting that the number of TFs targeted to the endoplasmic reticulum is probably low. The number of potential TFs with predicted transmembrane (TM) domains (Appendix A) was 24 (4.6%), according to the FungiDB database or the TMHMM website (a less restrictive prediction by Phobius suggested that the number of TM domain-containing TFs was 143; 27.5%) (Figure 1B). This suggests that there may be a significant subset of TFs participating in signal transduction mechanisms between membraneous compartments and nuclei; for example, this has been described for SrbA/AN7661, an SREBP-type (sterol regulatory element-binding protein) TF of *A. nidulans* [52]. Nevertheless, the number of active TM domain-containing TFs will in all probability be lower, according to the RNA-seq data that will be analyzed in more detail below.

### 3.2. Discrepancy between Genomic Annotation and Transcriptomic Data

In general, comparative analyses among species are conducted based on genome annotations. However, the reliability of eukaryotic genome annotations is considered low, due in part to the low percentage of eukaryotic genomes comprising protein-coding exons and that gene coding regions are interrupted by introns [53]. Thus, it was of interest what percentage of the potential *A. nidulans* TFome is transcribed as predicted, and if there was a subset of genes rendering TF forms different from the predicted full length forms or even non-functional. With this aim, RNA-seq experiments published by us previously [29,30,31] and transcriptomic data available in the FungiDB database were used as a reference to compare genomic annotations and the most probable transcript sequences (Figure 2).

The information of RNA-seq analysis for each predicted TF-coding gene is included in Appendix A (column B: RNA-seq data). It was not possible to assess the annotation of approximately 23 genes (out of 520; 4.4%; Figure 2A and Appendix A; the expression levels of *AN3391*, encoding a Zn2Cys6-type TF, are shown in Figure 2B as an example of this first group). The coding sequences of approximately 338 genes/transcripts are processed as predicted (65.0%; Figure 2A) in the culture conditions and genetic backgrounds analyzed (coverage of RNA-seq reads at the *AN2290/steA locus*, coding for a regulator of sexual development [54], is shown as an example in Figure 2C). In some cases, alternative splicing events were identified in transcripts, but the corresponding genes were included in this second group because the position and extension of the introns were correctly annotated. Intron retention (IR) was a commonly found alternative splicing form. The expression levels of *AN1569*, encoding a Zn2Cys6 TF, are shown as an example in Figure 2D. Retention of the first intronic (dotted purple squares) sequence leads to a premature stop codon that would generate a truncated peptide containing only the Zn2Cys6 IPD (amplification of the dotted orange square in the bottom panel of Figure 2D). Considering the low number of reads showing the processing of that first intron, it can be suggested that the concentration of the full length form of the protein will be, in all probability, very low.

Genes/transcripts needing some type of re-annotation were included in a third group (approximately 159 out of 520; 30.6%; Figure 2A). Of them, 124 were predicted to code for Zn2Cys6 and/or Fungal sp. IPDs, 17 for C2H2, 5 for bHLH, 3 for bZIP, 3 for homeodomain, 1 for a myb-like, 2 for GATA-type, 1 for Copper Fist, 1 for WOPR, 1 for Velvet and 1 for Forkhead domain IPDs. In multiple cases, the discrepancy between annotation and transcriptomic data was limited to the 5′- or 3′-end of the gene. This can be seen in the cases of *An11098*, encoding a Zn2Cys6-type TF, in which the start codon is probably located in the beginning of the predicted second exon (M23) (Figure 2E); and *AN2782*, also coding for a binuclear zinc cluster, in which the third predicted intron is absent (dotted purple square), modifying the reading frame and the stop signal (Figure 2F). Transcriptomic data also suggested that the coding sequences of a subset of genes within this third group clearly differ from the predicted ones. Three examples are shown in Figure 2. The start of the *AN1812/jlbA* transcript, which is upregulated under nitrogen starvation [55], is as annotated. However, the predicted first intron is absent and the 5′-UTR region of *AN1812/jlbA* is probably longer than predicted (Figure 2G). The probable start codon of the coding sequence will be that for Met175, with the bZIP domain predictably extending from residues 194 to 257 (the predicted polypeptide in FungiDB and AspGD databases is 258 amino acids long). *AN9221* is predicted to code for a TF with a signal peptide (SP) and six transmembrane (TM) domains in the N-terminus of the protein (first 249 amino acids), and a Zn2Cys6 IPD between residues 267 and 303 (Figure 2H). The number of reads mapping to the first two predicted exons and introns is very low compared to those mapping to the predicted third exon. RNA-seq reads also suggest that there is no intron in the sequence, and that *AN9221* is two separate genes, *alnG* and *alnR* of the asperlin cluster, as proposed by Grau and colleagues [56] (bottom panel in Figure 2H). The probable start codon of *alnR* is that which is annotated in the FungiDB database as Met246 for AN9221, just before the region coding for the binuclear zinc cluster domain. Finally, the analysis of *An5252* is shown in Figure 2I. Firstly, the predicted protein sequence of AN5252 is identical to that of AN9240, suggesting that they are not paralogs but rather correspond to a genome assembly/annotation error. Despite the low number of reads mapping to this gene (Figure 2I), RNA-seq data suggest that the gene contains only one intron (the predicted second intron). The 5′-UTR region is probably longer than predicted, and the real start codon possibly corresponds to that annotated as M117, implying the loss of the first predicted C2H2 domain (dark grey square bolded in red). The predicted second C2H2 IPD (M127-D168 plus 20 codons out of frame; coordinates of the FungiDB version) would be included in the coding region of AN5252.

### 3.3. A Subset of Genes Potentially Coding for TFs, Mainly within the Binuclear Zinc Cluster Family, Remains Silent

To analyze the expression patterns of the genes encoding the potential TFome of *A. nidulans*, we retrieved RNA-seq data from multiple transcriptomic comparisons carried out using *A. nidulans* as the reference system and publicly available in the FungiDB database or other repositories (see Materials and Methods). Appendix A includes FPKM and TPM data for each TF-coding transcript, which are ordered according to the IPD family that the corresponding protein belongs to (dual specificity TFs, those containing more than one type of IPD [50], were included in all the families they belong to, and, thus, repeated). First, heatmaps were built for each comparison. Clustering was avoided to check if there could be any expression pattern associated with a specific IPD family. Appendix A shows that it is not the case.

As an important number of the transcripts in Appendix A showed low or null expression, all FPKM (heatmaps 1–6 and 9–10) or TPM (heatmaps 7 and 8) values below 5 were removed from the table to build Figure 3 (indicated in gray). The presence of larger gray areas (FPKM/TPM < 5), principally in the sections corresponding to Zn2Cys6-Fungal sp. or aflatoxin regulatory IPD families, followed by some C2H2-type TFs, suggests that an important fraction (at least 53 out of 520; 10.2%) of the TFome of *A. nidulans* tends to remain silent under the tested experimental conditions or genetic backgrounds (others are silent in most but not all culture conditions/genetic backgrounds). For example, this is expected in the case of TF-coding genes of secondary metabolite gene clusters (SM clusters). Nevertheless, among those 53 genes that remain silent in all conditions/backgrounds analyzed, only 13 are predicted or shown to be members of SM clusters (*AN0533*, *AN1029*, *AN2025*, *AN3250*, *AN3391*, *AN6440*, *AN7076*, *AN7913*, *AN8103*, *AN8111*, *AN8509*, *AN8916* and *AN9236*; Appendix A; Figure 3). As described above, annotation of others, such as *AN5252*/*AN9240*, is clearly incorrect (see Figure 2 and Appendix A), suggesting that no functional TF forms are translated from some predicted genes in detectable levels.

Figure 3 and Appendix A also suggest that specific culture conditions and genetic backgrounds trigger a change in the general expression pattern of the *A. nidulans* TFome (see black bars in Figure 3 and Appendix A). The clustering of heatmaps corresponding to the effect of the addition of sugarcane bagasse as the main nutrient source (Figure 4A,C) [35], or a null *sltA* (salt sensitivity) background (Figure 4B,D) [29], supports this possibility. Qualitatively, a majority of TF-coding genes seem to be upregulated in the first culture conditions, but downregulated in the *sltA*Δ mutant.

### 3.4. Duplication of TF-Coding Genes Impacts the Species Level

In a previous work, and using BLAST and HMM analyses, the conservation pattern of 33 regulators of fungal development was analyzed in 503 fungal proteomes representing all phyla and subphyla, and most known classes [9]. The different conservation patterns observed suggested stepwise emergence of transcription factors. Interestingly, the conservation of a subset of those transcriptional regulators was restricted to the class and order levels, and in specific cases almost to the species level, as apparently occurred with MoConX4 of *Magnaporthe oryzae*. A similar trend was observed by Krizsán and colleagues [8]. Consequently, it was suggested that the emergence of some TFs may be recent, and may have led to the rewiring of the transcriptional networks controlling development and to further morphological diversification.

The FungiDB database includes an OrthoMCL identifier for each gene [57]. This preliminary analysis of orthologs and paralogs within the FungiDB database gave the opportunity to make the first screening of *A. nidulans* TFs with a restricted phylogenetic distribution. As the number of listed orthologs was very low for specific bHLH-type TFs and included additional *A. nidulans* bHLH-type TFs, the search was focused on this family of TFs. Thirteen genes predicted or shown to code for TFs of this family were identified by the FungiDB database. Five of them had a standard name (*AN1298/glcD*, *AN7553/devR*, *AN7661/srbA*, *AN7734/anbH1* and *AN8271/palcA* [52,58,59,60,61]), while the remaining eight had only a systematic name. By sequence similarity, AN4394/UrdA was initially described to belong to this family of TFs [31,62]. However, the Interpro website did not identify a basic helix-loop-helix domain within its primary structure. Thus, it was included in Appendix A, but excluded from the following analysis.

Firstly, to analyze the evolution of *A. nidulans* bHLH-s, sequences of proteins that were predicted to contain this type of transcriptional regulatory domain were retrieved from the FungiDB database. The search was limited to *Aspergillus* species included in this database, plus *Penicillium rubens*, *Talaromyces stipitatus* and *T. marneffei*. The phylogenetic tree (Figure 5A) suggested that all but four *A. nidulans* bHLH sequences (AN0396, AN2811, AN2392 and AN5078) are conserved in the family Trichocomaceae.

In the case of *AN5078*, RNA-seq data strongly suggest that the predicted two introns are not present, modifying the start codon to that in position 92 (Figure 5B). BLAST searches and HMMER analyses using this shorter sequence gave several possible orthologs of AN5078. Nevertheless, a reverse BLAST of each specific hit against the genome database of *A. nidulans* at the AspGD database (confirmatory reverse retrieval [9]) returned AN5078 as the first hit in very few cases (Appendix A), confirming a conservation pattern for AN5078 restricted to less than ten *Aspergillus* species.

The phylogenetic tree in Figure 5A included AN0396 in the AN5078 clade. RNA-seq data showed that the predicted intron of *AN0396* is processed, and the predicted protein sequence is in all probability correct (Figure 5C). BLAST and HMMER searches, as well as confirmatory reverse retrievals, showed conservation of AN0396 exclusively in *A. mulundensis*, with the rest of hits being apparently orthologs of AN5078 first, and AN7170 in the second instance. These results strongly suggest that AN0396 corresponds to an even more recent duplication of AN5078.

AN2392 is predicted to be a TF of the AN10297-containing (coding for a non-ribosomal peptide synthetase, NRPS-like protein) secondary metabolite gene cluster. Despite the low expression levels, mapping of RNA-seq reads suggest that the predicted second intron is not present, leading to an earlier stop codon prior to the last two predicted exons (Figure 5D). BLAST and HMMER searches, together with confirmatory reverse retrievals, strongly suggest that AN2392 corresponds to a duplication of AN6295 or AN10132, and that its conservation pattern is highly restricted, a common feature of genes belonging to secondary metabolite gene clusters [63,64]. Finally, despite the low expression levels, RNA-seq data suggested that the annotation of *AN2811* is incorrect and that the probable amino acid sequence is shorter than that which is predicted, but it probably includes a bHLH domain (Figure 5E). BLAST and HMMER searches gave 19 probable orthologs, and in all cases the confirmatory reverse retrieval against the *A. nidulans* genome at the AspGD database returned AN2811 as the first hit. The length of all these hits was in the range of 83 (*A. nidulans*) and 169 aa-s (*A. thermomutatus*). Thus, results suggest that AN2811 is conserved at least in the genus *Aspergillus*. Overall, the analysis of the bHLH family of TFs in *A. nidulans* suggests that this family is prone to duplication, and that the most recent duplication events impacted the species level (see Discussion).

### 3.5. The Zn2Cys6 Family of TFs as a Paradigm of Gene Duplication Events Extending the Transcriptional Regulatory Potential of A. nidulans

The biggest family of TFs in *A. nidulans* in particular, and Ascomycetes in general [50,51], is that of zinc clusters. In addition, gene duplication and the generation of paralogs is one of the main events causing rewiring of GRNs, and extension of the regulatory potential (see the Introduction and Discussion of this work). To update and extend the initial characterization of zinc clusters carried out by Todd and Andrianopoulos [65], the CD-Hit website (sequence identity cut-off of 0.3; see Materials and Methods) was used to search for paralogs within this family of TFs (Figure 6). CD-Hit identified clusters of up to six paralogous Zn2Cys6 TFs, totaling 139 members of this family that could have arisen by gene duplication. This number means that at least 42.0% of all predicted *A. nidulans* Zn2Cys6-type TFs would be a result of paralogy (potential clusters of 2, 3, 4, 5 and 6 paralogs are indicated in Figure 6A, with brown, green, purple, red and blue lines, respectively; circles indicate TFs predictably involved or shown to be involved in the control of secondary metabolism, while stars indicate those with a standard name). The biggest three clusters, composed of six paralogs each, were taken as a reference. Unexpectedly, the six members of one of them, AN7896/DbaA, AN2036, AN3385, AN0533, AN6788 and AN1029/AfoA, are all regulators of secondary metabolite gene clusters. A BLAST alignment, using AN3385 as query, shows the high score, expected and coverage values (Figure 6B; the sequence alignment can be seen in Appendix A; the current version of AfoA in the FungiDB database does not include the Zn2Cys6 domain; a previous version including it, XP_658633.1, was used in the alignment based on [56,66]). A similar pattern was observed in the remaining two clusters of six paralogs (Figure 6C,D; sequence alignments shown in Appendix A, respectively). In the case of Appendix A, two of the six Cys residues of the Zn2Cys6 domain of AN10906 seem to have mutated. Analysis of RNA-seq results strongly suggest that the annotation of the gene is correct. Thus, it would be of interest to assess how the divergent evolution of the Zn2Cys6 domain of AN10906 could affect its ability to bind DNA and its overall functionality.

Overall, the above results correlate with the idea of gene duplication being a main event in the extension of the regulatory potential of organisms, and suggest that duplication of genes coding for Zn2Cys6 TFs has played a key role in the expansion of this family of regulators in Ascomycota. Nevertheless, it remains to be investigated why this family of TFs has been expanded in this phylum, and not the others (see Discussion).

As a comparison, the same procedure was followed with C2H2-type TFs, the second biggest family of TFs in *A. nidulans*. The CD-Hit suite predicted paralog clusters of two or three members only (1: AN0486, AN2270 and AN9492; 2: AN0885, AN9328 and AN10334; 3: AN6733 and AN11197; 4: AN0096 and AN7118; 5: AN0644 and AN2498; 6: AN1997 and AN5583; 7: AN5929 and AN9017). A total of 16 (considering that AN5252 and AN9240 are the same protein; see above) out of 73 potential C2H2-type TFs were predicted to be a result of duplication, a clearly lower percentage (21.9%) compared to that estimated in the case of Zn2Cys6 TFs (42.0%).

The case of the developmental regulator BrlA [67] is paradigmatic among C2H2-type TFs of *A. nidulans*, since it is conserved exclusively in the order Eurotiales and maybe in Chaetothyriales (Eurotiomycetes), but occupies a central position in the genetic pathways that control conidiation [9,12,13]. Indeed, BrlA links the upstream developmental activation (signal transduction) pathways (UDA), and the central developmental pathway (CDP) that controls most of the morphological changes that lead to conidia production [49]. Interestingly, the promoter of *brlA* and its gene products developed the ability to recruit upstream TFs and transcriptionally control downstream TF-coding genes that are more widely conserved in fungi. Thus, it constitutes a model system for the study of how the emergence of new regulators can trigger GRN rewiring [9]. BLAST and HMM analyses carried out to determine the conservation pattern of BrlA in the proteomes of 503 fungal species [9] returned hits for BrlA in species outside Eurotiomycetes. Nevertheless, according to the confirmatory reverse retrieval against the *A. nidulans* proteome within the AspGD database, these hits corresponded, in most of the cases, to orthologs of SteA (not shown), a TF widely conserved in fungi and necessary for sexual development (Figure 2C) [54].

In an attempt to obtain clues on the origin and the ancestor(s) of BrlA, I compared the C2H2 IPDs of BrlA with those of specific *Aspergillus nidulans* and *Saccharomyces cerevisiae* TFs of the same family (Figure 7A). The sequences aligned in Figure 7A conserved all but one of the residues of *S. cerevisiae* Msn paralogs (cerMsn2 and cerMsn4), predicted to contact DNA (green and red arrows, respectively) [68]. The only divergence was the substitution of a Gln in the second C2H2 IPD of Msn proteins, and SteA by an Ala in BrlA. Thus, it is tempting to suggest that the C2H2-type transcriptional regulatory region of the asexual developmental regulator BrlA belongs to the same family of those of MsnA and SteA, which are regulators of stress-response and sexual development, respectively. However, as expected, the C2H2 domains of MsnA (residues T463-H515) could not functionally replace those of BrlA (residues F320-H375), as a recombinant strain expressing a BrlA::HA_3x_ chimera bearing this replacement (M1-Q319 of BrlA, T463-H515 of MsnA, and S376-E432 of BrlA) showed an aconidial phenotype (Figure 7B).

### 3.6. Identification of a Zn2Cys6-Type Protein Necessary for Growth and Development

To understand the structure, function and evolution of complex GRNs, scientists hierarchically decompose these networks into manageable and intelligible subsystems (see references in [69]). This approach has been informative, going back again to the example of the control of asexual development in *A. nidulans*, with UDA and CDP pathways and the link established between them by BrlA (see the previous section of this work). Nevertheless, in general, GRNs are connected with other networks. Indeed, TFs inducing sexual development, such as NsdD, act as repressors of asexual development by directly binding to the promoter of *brlA* and inhibiting its expression [70,71]. In addition, the expression of *brlA* is induced or repressed, depending on several environmental stimuli, such as light/darkness, the presence of O_2_/CO_2_, salt or osmotic stress conditions or nutrient saturation/starvation (see references in [49]). For example, the emergence of hyphae to the air environment induces the expression of *brlA* through the UDA pathway, but at the same time, nitrogen starvation also induces the expression of *brlA* and conidia production in submerged culture [72].

Sibthorp and colleagues analyzed the effect of the nitrogen source (ammonium or nitrate) added to the culture medium, or the effect of nitrogen starvation (4 and 72 h) in the expression profiles of *A. nidulans* [27]. RNA-seq data of this experiment were based on the SOLiD (sequencing by oligonucleotide ligation and detection) sequencing technology and corresponded to a single experiment without biological replicates. Thus, they were not intended for use in comparative expression analyses, but to provide expression information to assist genome annotation [27,73]. This means that quantitative expression data must be handled with care. Nevertheless, as the transcriptional responses to nitrogen starvation and conidiation may overlap partially, I hypothesized that FPKM values provided by the FungiDB database could be used as preliminary data to explore a hypothetic relationship between nitrogen starvation and the induction of development. For example, when comparing the sample collected under nitrogen starvation (72 h) and that in AMM that used ammonium as the nitrogen source (Appendix A), most of the genes seemed to be upregulated (2219 genes with a FC ≥ 3, compared to 781 genes with a FC ≤ 3; ratio ~ 2.8; black squares in Appendix A). Genes encoding TFs (Zn2Cys6-fungal specific and C2H2 families) were apparently among the top Interpro domains upregulated under nitrogen starvation (196 TF-coding genes were upregulated and 18 downregulated; ratio ~ 10.9; Appendix A; Appendix A). As expected, genes encoding key regulators of nitrogen metabolism were found in the list of upregulated TF-coding genes, but also several known regulators of asexual and sexual development (Appendix A) [31,59,67,74,75,76,77,78,79]. Thus, it was hypothesized that new developmental regulators could be identified in this preliminary list. Five genes were selected (Appendix A) and their null mutants generated.

Multiple criteria were defined for the selection of these five genes. First, orthologs of genes that in other fungi, such as *Neurospora crassa*, are important for development were included (e.g., according to the FungiDB database, *An10192* is the ortholog of *N. crassa tah-2*). Second, genes significantly deregulated after the induction of asexual development were included (*AN1500* and *AN4324*, respectively) [30]. Third, genes with one (*AN10192*) or no paralogs (*AN1500*, *AN2001*, *AN4324* and *AN4527*) were prioritized. Finally, genes coding for TFs of the most numerous families were selected (*AN10192* and *AN2001* encode zinc clusters, *AN1500* encodes a C2H2-type TF, *AN4324* encodes a bZIP-type TF, and *AN4527* encodes a cMyb-type TF). In comparison with the parental wild-type and the remaining null mutant strains generated, the *AN2001*Δ mutant showed partial inhibition of radial growth and conidia production, mainly in ACM (*Aspergillus* complete medium) supplemented with nitrate as the main nitrogen source (Figure 8A,C). After 144 h of culture at 37 °C, while the reference wild-type strain produced cleistothecia in ACM (Figure 8B,D), the *AN2001*Δ mutant was blocked for the development of sexual structures.

Overall, results strongly suggest that in the future, the analysis of the transcriptomic responses of *A. nidulans* to nitrogen starvation could help in the identification of hitherto uncharacterized regulators of development. However, such analysis should be based on RNA-seq data that include biological replicates. In addition, considering that nitrogen starvation or the use of different nitrogen sources can acidify (i.e., ammonium) or alkalinize (i.e., nitrate) the culture medium, it can be hypothesized that specific regulators of nitrogen metabolism and/or stress-response could act indirectly or directly by binding to the promoter of *brlA* as developmental activators. In the case of AN2001, our group plans to elucidate the functional relationship with known regulators of growth and sexual/asexual development, as well as the main features of its regulatory activity. Currently, it can be stated that it shows moderate/high expression levels under most of the conditions and in most of the genetic backgrounds tested through RNA-seq in *A. nidulans* (Appendix A), its three introns are processed as predicted (Appendix A) and it is conserved in the subphylum Pezizomycotina (not shown).

## 4. Discussion

### How GRNs Evolve and What Fungi Can Teach Us about the Process

There are three main properties that help us understand how GRNs evolve (see references in [80]). Robustness is related to the ability of a system to sustain its functionality in the face of perturbation [81,82], which, in the case of GRNs, could be understood as the extent a phenotypic trait persists after mutation [80]. Tunability can be defined as the changes in gene expression levels, for example, due to gain, loss or mutation of CRSs, or mutation of the DBDs of TFs. Finally, evolvability, beyond the generic ability to evolve or the capacity of an evolving system to generate or facilitate adaptive change, is associated with the capacity of GRNs to modify or acquire novel regulatory connections [69,80]. Gene duplication and divergent sequence evolution of paralogs are central enablers for these three parameters, and, thus, for the rewiring and reorganization of GRNs. As mentioned in the introduction of this work, duplication and sequence divergence will probably cause gains or losses of CRSs, and neofunctionalization or subfunctionalization would modify connections within or among gene networks [3,4,5]. Consequently, all of these events could directly impact robustness, tunability and evolvability of GRNs.

Duplication of TF-coding genes and their divergent sequence evolution is also a main mechanism for GRN evolution in fungi. The availability of hundreds of fungal genomes and proteomes, representing most of the clades of the fungal tree of life [7,83], gives fungal researchers the possibility to elucidate when each family of transcriptional regulators and each TF emerged [8,9,51,84,85]. *A. nidulans* has been used as a reference fungus since the 1940s. The major transcriptional regulators of multiple GRNs, such as those controlling sexual and asexual development, stress-response, carbon and nitrogen metabolism, secondary metabolism, and so on, have been functionally characterized in *A. nidulans*, making it a valuable system for the study of how GRNs are rewired and evolve after the emergence and duplication of new regulators.

Knowledge gained can be extended to other fungal systems. Recently, new experimental approaches have been developed, focusing on the expression of genes encoding regulators of *A. nidulans* asexual reproduction with the aim of modifying developmental patterns of species such as *Monascus ruber* or *Histoplasma capsulatum* [12,86]. Although the trials were unable to trigger major morphological changes, they paved the way for the establishment of simple and reliable approaches for the experimental study of GRN rewiring and its consequences for organism evolution.

Tracking the emergence of fungal TFs has shown us that it occurs gradually, with specific clades exhibiting a burst in the number of new regulators [8,9]. Furthermore, results support the view that this process is currently active, with the existence of TFs with an extremely narrow or almost species-specific distribution. In plants, bHLHs form the second largest family of TFs, where they are key regulators of important metabolic, physiological and developmental processes [87]. This family of TFs had been shown to be remarkably expanded approximately coincident with the appearance of the first land plants, but has subsequently remained relatively conserved in this group of plants [87,88]. In mollusks, Bao and colleagues proposed that bHLH genes showed high evolutionary stasis at the family level, but considerable within-family diversification by tandem gene duplication (TGD) [89]. The results obtained here from the analysis of the set of bHLH-type TFs of *A. nidulans* correlate with the idea of high within-family diversification, though not through a tandem gene duplication (see below for Zn2Cys6-type TFs). Thus, members of this family of TFs could be viewed as a paradigm of intra-genus gene duplication. *AN2392* probably emerged as a duplication of *AN6295* or *AN10132*. *AN0396* seems to be an even more recent duplication of *AN5078*.

However, the largest family of TFs in *A. nidulans* in particular, and Ascomycetes in general, is that of Zn2Cys6-type TFs [50,51], occurring to a large extent due to multiple events of gene duplication. Shelest described that among the most abundant TF-coding gene families, not all of them expand, not all of those expanding do so at the same rate and not all of them expand following comparable trends in all fungal lineages [50]. Zinc clusters grew much faster in Ascomycetes, suggesting a particular role and evolutionary history of Zn2Cys6-type TFs [50]. An interesting question lies in what makes this family so prone to successful gene duplication in Ascomycetes (the same question could be made for C2H2-type TFs in Basidiomycetes and early diverging fungi).

Gene duplication can occur by several mechanisms, including whole- or segmental- genome duplication, and single gene duplication, which includes tandem, proximal, retro-transposed, DNA-transposed and dispersed duplications [90,91]. Whole-genome duplication events have been described in fungi [92], but not in *Aspergilli* [93]. Appendix A and Appendix A show the genomic distribution of those genes encoding the Zn2Cys6-type TFs shown in Figure 6A, which could constitute clusters of paralogs according to the CD-Hit algorithm. It has been described that specific mechanisms of TGD might lead to increasing or decreasing copy numbers in gene families [90]. However, based on their chromosomal distribution, TGD seems to not be the main mechanism of gene duplication of those probable paralog clusters of Zn2Cys6 TFs of *A. nidulans*. Segmental duplications (SDs) are defined as genomic segments longer than 1 Kb, that are repeated within the genome with at least 90% sequence identity [94]. However, as mentioned previously, Khaldi and Wolfe described that they found no clues of SDs in *Aspergilli* [93]. Dispersed duplications (DDs) have been defined as the generation of two gene copies that are neither neighboring nor colinear [95], a definition which fits with the genomic distribution of the members of the hypothetical Zn2Cys6 paralog clusters of *A. nidulans* (Figure 6A and Appendix A, Appendix A). Despite the fact that the mechanisms enabling DDs remain unclear and have been defined as unpredictable and occurring through random patterns [95], dispersed duplication can be proposed as one of the main duplication events causing the expansion of Zn2Cys6-type TFs in *A. nidulans*.

Interestingly, there are Zn2Cys6 paralog clusters that could have been generated following different duplication mechanisms. One of the biggest Zn2Cys6 paralog clusters is composed of regulators of secondary metabolite gene clusters (AN7896/DbaA, AN2036, AN3385, AN0533, AN6788 and AN1029/AfoA; Figure 6B). A BLAST search against the genome of *A. nidulans* using the sequence of AN7903/pkeA (the polyketide synthase, PKS, of the *dba* cluster) as the query [96], suggested that the PKSs AN2032/PkhA, AN3386/PkiA, AN0523/PkdA and AN1034/AfoE (the expect, score and coverage values of AN6791 were considerably lower) could also be paralogs (these were the best hits at least). Similar correlations were observed in the case of the esterases/lipases AN7899/DbaE, AN2031, AN0529, AN6789 and AN1032/AfoC, or the FAD-binding monoxygenases AN7902/DbaH, AN2033, AN3382/SalA, AN0530 and AN1033/AfoD. Additional BLAST searches against the *A. nidulans* genome also related AN7893 with AN0526, AN11584/DbaC (located between AN7897/DbaB and AN7898/DbaD) with AN0527, or AN7898/DbaB with AN0528. It has been proposed that the presence of the *dba* cluster in *A. nidulans* is a consequence of a horizontal gene transfer (HGT) event from *Talaromyces stipitatus* [31,97], while the donor of the cluster defined by AN2036 was probably *Coccidioides immitis* [31] or *Penicillium italicum* [97]. The rest of the secondary metabolite gene clusters corresponding to the aforementioned Zn2Cys6 paralog group have not been linked with HGT events [97], and could correspond to internal duplication events (followed by divergent evolution). Again, the location of these clusters in different chromosomes makes the possibility of TGD events unlikely. Furthermore, the presence in the cluster defined by the PKS AN0523/PkdA of a DDE1 transposon-related protein, AN0532, predictably containing a Tc5 transposase DNA-binding domain (IPR006600), a DDE superfamily endonuclease domain (IPR004875) and a Homeodomain-like domain (IPR009057), renders duplication via the action of transposable elements plausible. Overall, quantification of the importance of each duplication mechanism in the expansion of Zn2Cys6-type TFs in the *A. nidulans* genome is beyond the scope of this work and will require deeper bioinformatic analyses; however, these observations support that DDs and transposon-mediated duplication mechanisms are more relevant than the other mechanisms.

The most likely possibility after duplication is the loss of the new paralog. If the duplication event persists in the population, then pseudogenization would be a probable outcome [90]. The pseudogene would then be free to accumulate deleterious mutations, as could be the case in *AN0737* or *AN1887* (which are probably duplications of *AN1518* and *AN10300*, respectively; pseudogenes were removed from the initial list of TF-coding genes in Appendix A). However, multiple duplications are fixed and maintained in genomes, and different models have been proposed to explain how neofunctionalization, subfunctionalization or functional redundancy occur [90,98]. Examples of neo-/subfunctionalization, or functional redundancy within Zn2Cys6 paralog clusters could be those of AN7610/XlnR (xylan degradation), AN0388/AraR (pentose catabolism) and AN10550/GalR (galactose conversion), which are involved in the control of carbohydrate metabolism [99,100] and form a paralog cluster with AN9458 and AN2672. Also the cases of AN1848/RosA and AN5170/NosA, both involved in the control of sexual development [101,102]. A third example could be the paralog cluster composed of AN1425/FarB and AN7050/FarA (fatty-acid utilization) [103], together with AN9117, AN3863 and AN1077. It has been proposed that the variability in the general structure of CRSs bound by Zn2Cys6 TF-s at target promoters could be a main reason supporting their fixation and expansion in the genomes of Ascomycetes [50]. This variability would allow highly specific recognition, but at the same time flexibility to produce a large number of different variants. Nevertheless, the fixation of paralog genes in a genome should also include the generation of CRSs in their own promoter regions, which would be bound by upstream TFs, in order to induce their expression, or, alternatively, by TFs with an autoregulatory activity, as could be the cases of AlcR, CpcA, FacB, PacC and others. In this scenario, Figure 3 and Appendix A indicate that a large proportion of genes encoding Zn2Cys6-type TFs show very low expression levels or remain silent. This includes paralog clusters in which some paralogs are expressed while others remain silent (Appendix A). It remains to be determined if this observation reflects the requirement of the corresponding functions under specific environmental/culture conditions (as could be the case with regulators of secondary metabolite gene clusters or that of specific TFs regulating their own expression), or, alternatively, that these genes are in a post-duplication stage (1) prior to gains of up- and/or downstream CRSs (fixation) or (2) prior to pseudogenization. In the future, the potential of silent TF-coding genes for GRN rewiring will need to be assessed by comparing the phenotypes of their null or overexpression mutants and by assessing which regulatory connections within GRNs are altered.

The second largest family is that of C2H2-type TFs, which includes several regulators of sexual and asexual development and stress-response, such as SteA, MsnA, BrlA, CrzA, PacC, FlbC, SltA, NsdC, AslA or RocA. BrlA, SteA and MsnA are remarkable due to their function in the control of asexual development, sexual development and stress-response, respectively, as well as the high conservation within the C2H2 domains of the amino acids that are predicted to bind DNA [68]. Although the high sequence divergence of these three regulators outside the DBDs will in all probability introduce specificities for DNA binding, it could be hypothesized that the DNA target sequence of BrlA could be similar to the consensus target of yeast Msn-like TFs (5′-A/CGGGG-3′) [68]. As can be seen, similarities compared to BrlA response elements (BREs; 5′-CAAGGGG-3′), defined by [104] based on the promoter of *rodA*, are remarkable, although it is unknown if this sequence is the consensus bound by BrlA in all of its target promoters. A plausible possibility is that BrlA, SteA and MsnA compete for the same DNA targets in specific promoters. On the one hand, in correlation with this hypothesis are the increase in conidia production of the null *msnA/nrdA* strain and the aconidial phenotype of overexpression and multicopy strains (see references within [105]). On the other hand, the *fluffy* phenotype described after the replacement of the C2H2 domains of BrlA with those of MsnA suggests that there are specificities in CRSs. Protein–DNA binding experiments with BrlA and MsnA constructs in wild-type and the corresponding null genetic backgrounds would render important information on (1) the evolutionary origin of BrlA, (2) its DNA-binding mechanism, and (3) how gene duplication can trigger rewiring of transcriptional networks and the modification of homeostasis/developmental responses. These strategies would contribute to the consolidation of filamentous fungi as reference systems to study GRN rewiring mechanisms.

## Figures and Tables

**Figure 1 jof-07-00600-f001:**
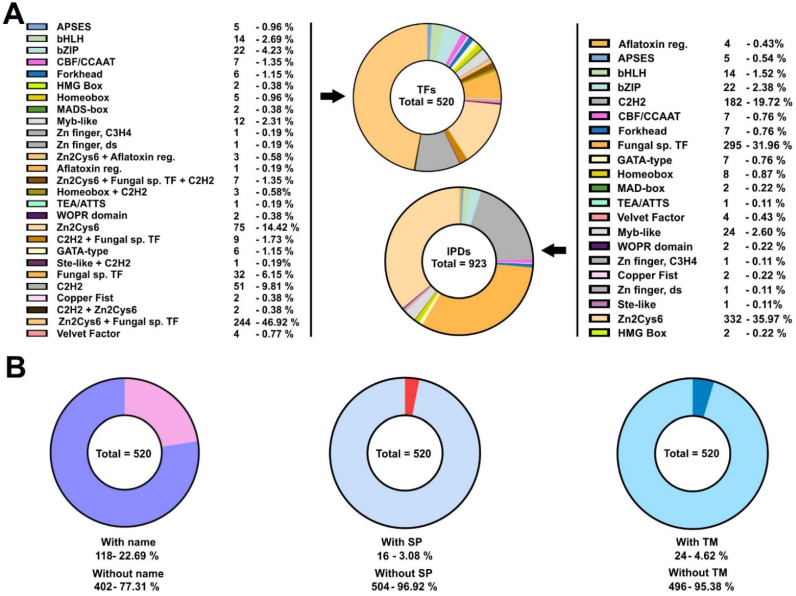
Potential TFome of the model filamentous ascomycete *A. nidulans*: (**A**) Pie charts showing the number and percentage of TFs (left; of a total of 520) and IPDs (right; of a total of 923) of each family potentially encoded by the genome of *A. nidulans*. Color keys are included. Both tables are an update of the list published by Wortman and colleagues [14]. (**B**) Pie charts showing the number of genes coding for TFs with a standard name (left), the number of potential TFs predicted to include a signal peptide in their sequences (middle) or those predicted to contain TM domains (right). See also Appendix A.

**Figure 2 jof-07-00600-f002:**
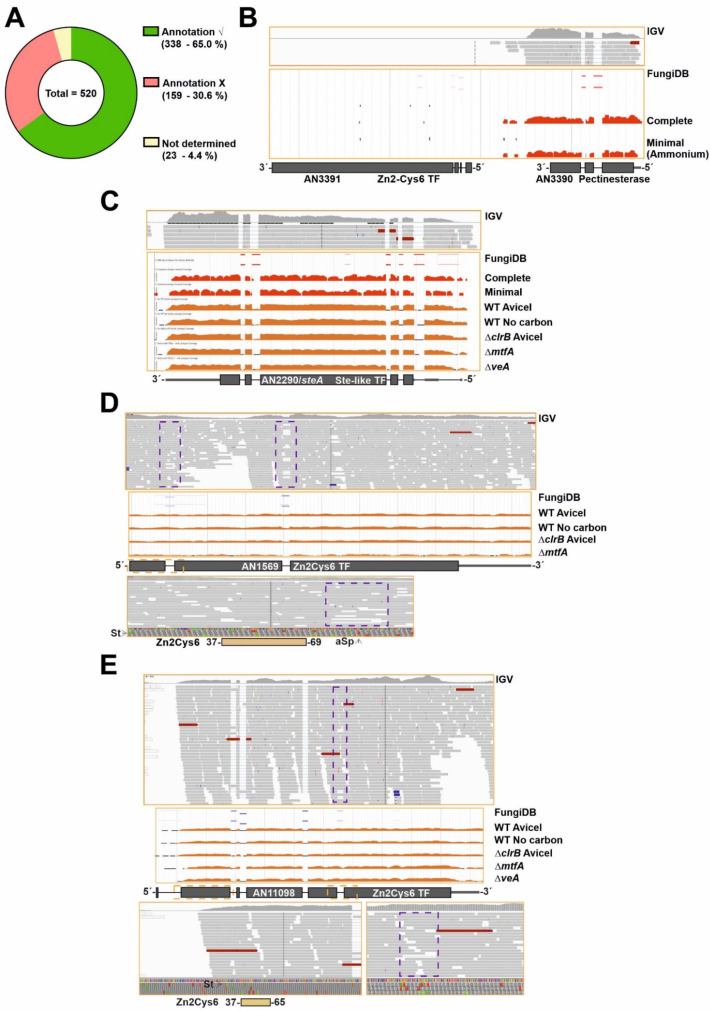
Comparison of annotation of genes potentially encoding TFs and RNA-seq data. (**A**) Pie chart showing the approximate number and percentage of TF-coding genes correctly annotated according to RNA-seq data (green), the fraction of those that should be less or more extensively re-annotated (red) and the fraction of those that could not be assessed (mostly due to their very low or null expression levels) (yellow). Approximately 30.6% of the TF-coding genes would need some re-annotation. (**B**) Null expression levels of *AN3391*, predicted to encode a C2H2- plus a Zn2Cys6-type dual-specificity TF, in comparison to those of *AN3390*, encoding a putative pectinesterase, as an example of genes in which the annotation could not be assessed. Data from our RNA-seq experiments (visualized with IGV; [29,30,31]) and those provided by the FungiDB database were analyzed [27]. (**C**) Expression levels of *AN2290/steA* as an example of a gene correctly annotated. RNA-seq experiments retrieved from the FungiDB database corresponded to complete and minimal media [27]; carbon starvation and a null *clrB* mutant [26]; and null *mtfA* or null *veA* backgrounds [28]. (**D**) Alternative splicing event, represented by the case of *AN1569*. Annotation of the coding region is correct but there are intron-retention (IR) events in both predicted introns (dotted purple squares); furthermore, the non-processed form is the most abundant. Retention of the first intronic sequence leads to a premature stop codon that would generate a peptide containing only the Zn2Cys6 IPD (amplification of the dotted orange square in the bottom panel). The concentration of the full-length form will be, in all probability, very low. (**E**,**F**) Examples of genes whose coding sequences should be re-annotated in 5′ (*AN11098* in panel **E**) and 3′ (*AN2782* in panel **F**) ends. The dotted orange squares indicate the regions amplified in bottom panels while the dotted purple squares indicate an intron retention event (**E**) or the absence of an intron (**F**). The non-processed form of the fifth predicted intron of *AN11098* adds, compared to the annotated sequence, 17 extra codons between those coding for amino acids D399 and G400, but without modifying the reading frame. (**G**–**I**) Examples of genes which would need major re-annotation of their coding sequences. The first predicted intron of *AN1812/jlbA* (**G**) is not present, generating a long 5′-UTR region. The probable start codon will be that for Met175, retaining only the sequence encoding the bZIP domain. The predicted two introns of *AN9221* (**H**) are probably absent, generating two coding sequences, *alnG* and *alnR* of the asperlin cluster (see main text). The coding region of *alnR*, a TF of the asperlin metabolic cluster, will probably begin with the codon for Met246 (FungiDB coordinates). The first predicted intron of *AN5252* (or *An9240*) (**I**) is not present, possibly leading to two coding sequences (those covering M1-G90 plus 9 codons and M127-D168 plus 20 codons). The first C2H2 domain (dark grey square bold in red) would be lost, while the second coding region would include the second C2H2 IPD. St: start codon. pSt: predicted start codon. Sp: stop codon. pSp: predicted stop codon. aSp: alternative stop codon. RF: reading frame. See also Appendix A.

**Figure 3 jof-07-00600-f003:**
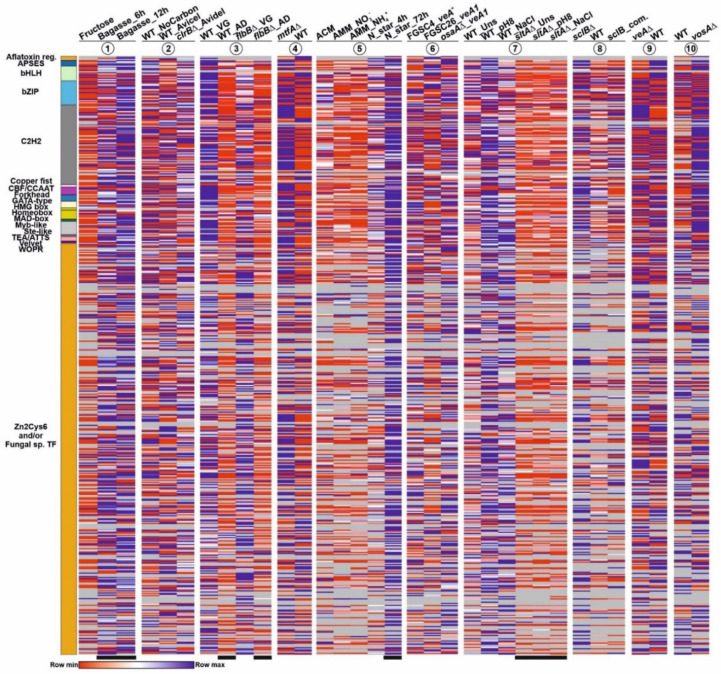
Expression pattern of each TF-coding gene of each IPD family under different culture conditions and in genetic backgrounds. Heatmaps were not clustered, so as to observe if there could be any general expression pattern associated to a specific family of IPDs. Colors in the left bar indicate each IPD family and follow the same color key as in the previous figures and tables. The extension of each color bar reflects the predicted number of genes within each family of IPDs (each row in the heatmaps). Heatmaps 1 to 10 show variation of expression (1) with the use of sugarcane bagasse (FPKM values), (2) in a null *clrB* mutant (FPKM), (3) a null *flbB* mutant (FPKM) before (VG) and after (AD) the induction of conidiation, (4) a null *mtfA* background (FPKM), (5) nitrogen starvation (FPKM; see main text), (6) a null *osaA* background (FPKM), (7) a null *sltA* background (under standard culture conditions, pH 8, or addition of 1M NaCl) (TPM), (8) a null *sclB* background (TPM), (9) a null *veA* background (FPKM) and (10) a null *vosA* background (FPKM). See references in the Materials and Methods section. All FPKM/TPM values below 5 were removed from the heatmaps and are indicated in gray, in order to highlight those TF-coding genes showing low or null expression levels in most or all genetic backgrounds and culture conditions. Black bars below the heatmaps indicate genetic backgrounds and culture conditions in which a change in the general expression pattern of the *A. nidulans* TFome might be induced. See also Appendix A.

**Figure 4 jof-07-00600-f004:**
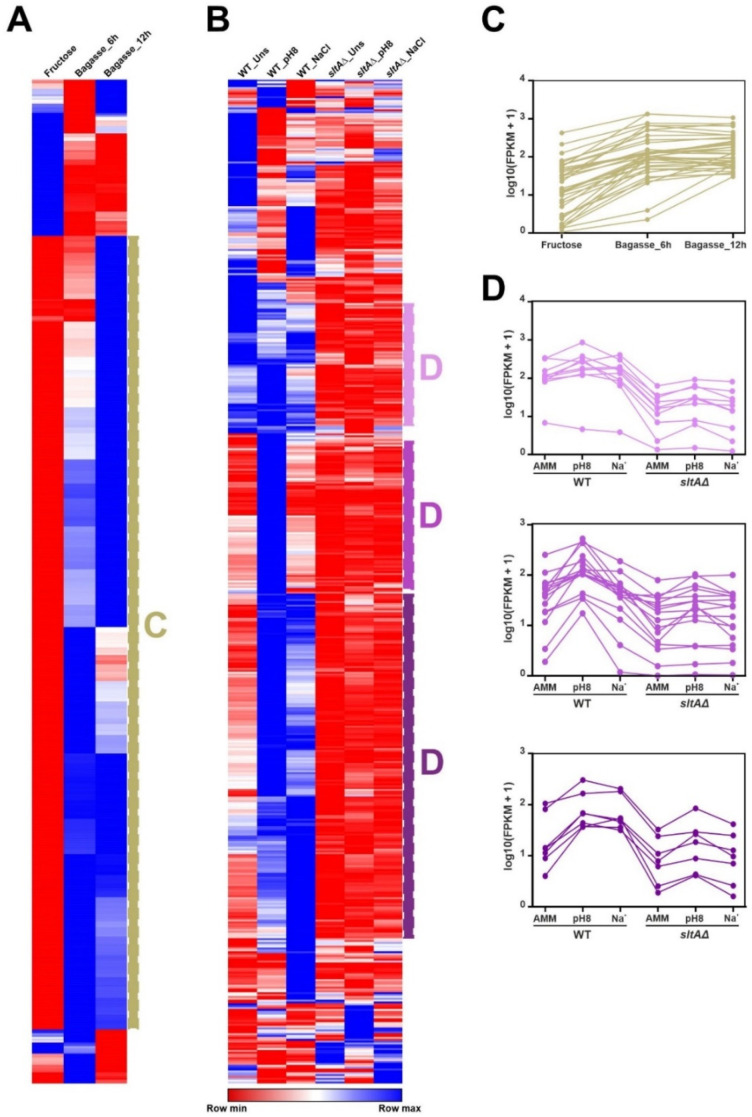
General modification of the expression pattern of TF-coding genes in response to specific culture conditions or in specific mutant backgrounds. Clustered heatmaps (Pearson correlation) showing a general, qualitative change in the expression levels of a majority of TF-coding genes in response to (**A**) the addition of sugarcane bagasse as the main nutrient source, or (**B**) deletion of *sltA*. Panels C and D show expression profiles of specific genes, selected from the groups highlighted in panels **A** and **B**, respectively. (**C**) Upregulation in a medium containing sugarcane bagasse. (**D**) Downregulation in a null *sltA* background in the three conditions tested (standard minimal medium or *unstressed*, alkalinity and sodium stress; upper graph), downregulation only at pH 8 (middle) or downregulation both at pH 8 and sodium stress (bottom).

**Figure 5 jof-07-00600-f005:**
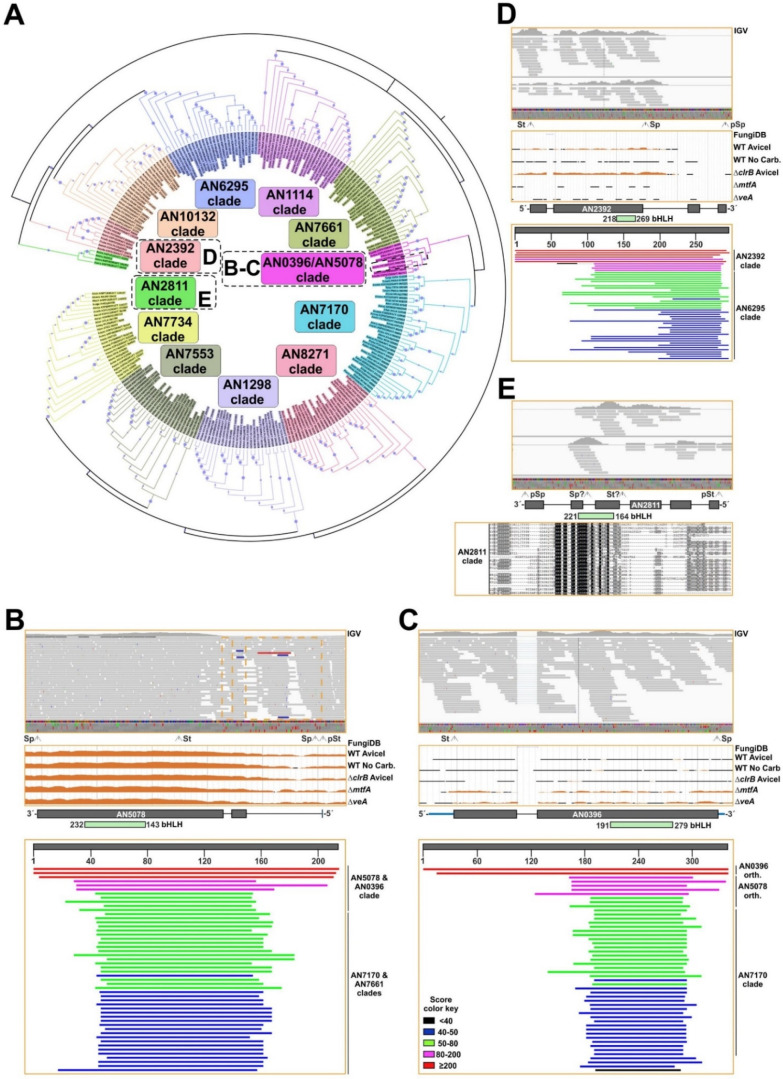
Evolution of the bHLH-type TF family of *A. nidulans*. (**A**) Phylogenetic tree showing the conservation of bHLH-type TFs of *A. nidulans* in the family Trichocomaceae. Sequences of predicted bHLH-type TFs in multiple *Aspergilli*, *Penicillium rubens*, *Talaromyces stipitatus* and *T. marneffei* were retrieved from the FungiDB database and aligned using Clustal Omega. The tree was generated using Mega, version 7.0.26 (maximum likelihood method and JTT model, with 60 bootstrap replications), and edited using iTOL (see Materials and Methods). The dotted black square in the AN0396 and AN5078 clade highlights the proximity of both sequences. Bootstrap values are indicated by blue circles. The accession number of each sequence is preceded by a code indicating the name of the species it belongs to (for example, Anidu indicates *A. nidulans*). (**B**–**E**) Density of reads mapping to *AN5078* (**B**), *AN0396* (**C**), *AN2392* (**D**) and *AN2811* (**E**) *loci*. Images were exported using the IGV software (top [29]) or from the JBrowse genome browser linked to the FungiDB database (middle). Bottom pictures correspond to BLAST searches from the NCBI website (panels **B**–**D**) or a Clustal alignment visualized with Genedoc (panel **E**). The positions of predicted introns and exons, predicted (pSt/pSp) and probable (St/Sp; based on RNA-seq data) start/stop codons, and the regions encoding the bHLH domains are indicated. See HMMER analyses in Appendix A.

**Figure 6 jof-07-00600-f006:**
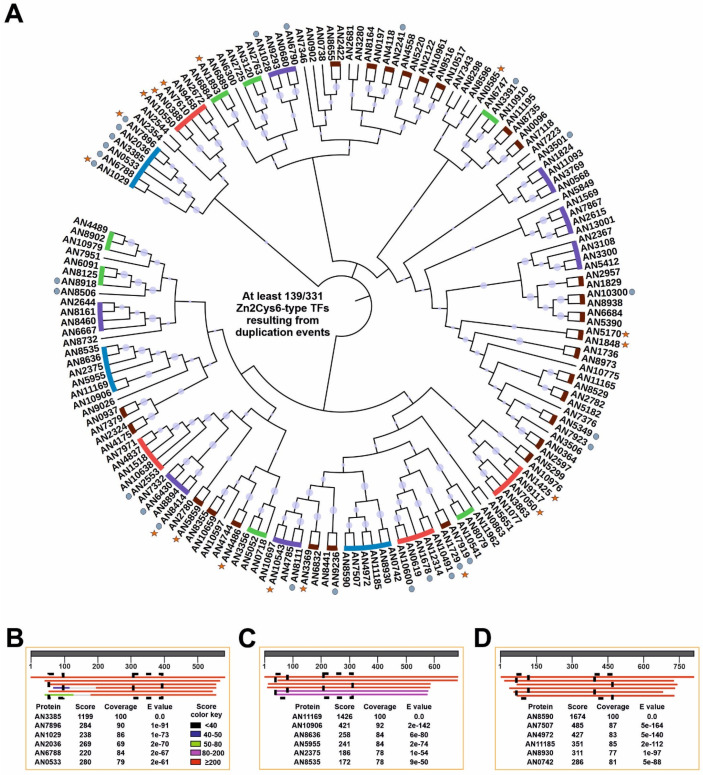
Duplication events within the zinc cluster family of TFs in *A. nidulans*. (**A**) Phylogenetic tree generated with Mega, version 7.0.26 (neighbor-joining method and 5,000 bootstrap replications), and edited using iTOL, showing the paralog clusters predicted by the CD-Hit suite. Predicted paralog clusters of 2, 3, 4, 5 and 6 members are highlighted with brown, green, purple, red and blue lines, respectively. Some Zn2Cys6-type TFs excluded from these paralog clusters (clusters of 1 member in the CD-Hit suite) were also included. Gray circles indicate those TFs predictably involved or shown to be involved in the control of secondary metabolism, while stars indicate TFs having a standard name. Bootstrap values are indicated by light purple circles. (B-D) BLAST alignments (NCBI website) for each of the three paralog clusters of six members identified by CD-Hit, using as queries AN3385 (**B**), AN11169 (**C**) and AN8590 (**D**), respectively. Score, expect and coverage values are included. The score color key is shown in panel B. The extension of Zn2Cys6 and fungal specific regulatory domains is shown in each panel with the black squares. The alignments of the sequences in each panel (**B**–**D**) can be seen in Appendix A, respectively.

**Figure 7 jof-07-00600-f007:**
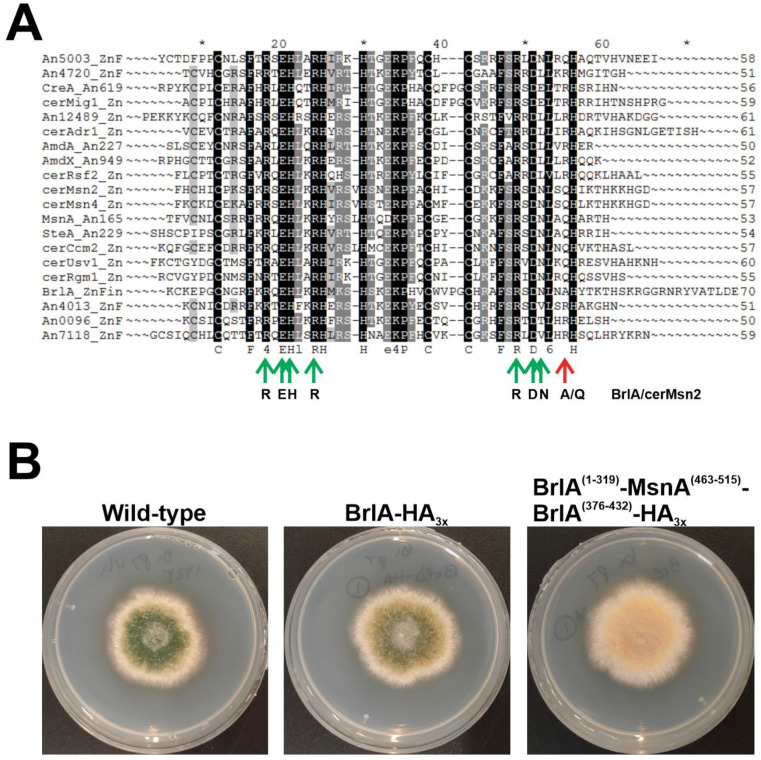
Similarities of the C2H2 domain of BrlA with those of specific regulators of sexual development and stress-response. (**A**) Alignment of the C2H2 DBDs of specific *A. nidulans* and *S. cerevisiae* (cer) TFs. The position and nature of the predicted DNA-binding amino acids (based on [68]) are indicated in green (conserved in all sequences analyzed, including BrlA) or red (not conserved in BrlA). The picture was exported from Genedoc, version 2.7.000, and edited. (**B**) Phenotype of a strain expressing a BrlA::HA_3x_ chimera in which the C2H2 domain of BrlA (residues F320-H375) had been replaced by that of MsnA (residues T463-H515), after 72 hours of culture at 37 °C in adequately supplemented AMM, and compared to the parental wild-type strain and a strain expressing BrlA::HA_3x_. The diameter of plates is 5.5 cm.

**Figure 8 jof-07-00600-f008:**
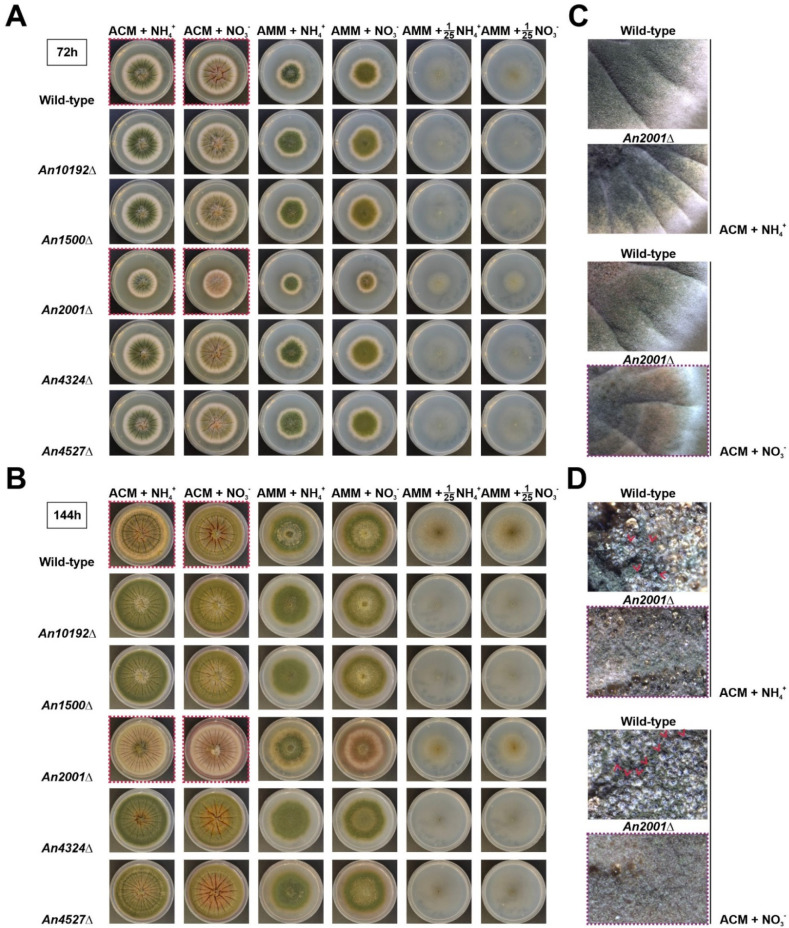
Phenotypes of single-null mutant strains of five TF-coding genes potentially necessary for *A. nidulans* development. Phenotypes of the null mutants in ACM (*Aspergillus* complete medium) or AMM supplemented with ammonium (first and third columns, respectively) or nitrate (second and fourth columns, respectively), after 72 (**A**) or 144 h (**B**) of culture at 37 °C, and compared to the parental wild-type strain. The phenotypes of the same strains under nitrogen limitation (1/25 dilution of the nitrogen source compared to AMM) are shown in columns five and six, respectively. The diameter of plates is 5.5 cm. Red squares indicate the colonies amplified in the images shown in panels (**C**) (72 h of culture) and (**D**) (144 h). Red arrowheads highlight cleistothecia.

## Data Availability

The data underlying this article are available in Appendix A. RNA-seq, genomic and proteomic data used in this article (and previously deposited by others under accession numbers PRJNA294437, GSE44100, SRX206691, SRX286239, SRP055436, PRJEB4484, GSE72316, PRJNA625291, E-MTAB-6996 and PRJNA588808) were retrieved from sources (www.ebi.ac.uk/arrayexpress/; www.ncbi.nlm.nih.gov/bioproject/; https://www.ncbi.nlm.nih.gov/geo/, accessed on 1 November 2020) and databases (https://fungidb.org/fungidb/app, accessed on 1 November 2020) in the public domain. *Aspergillus nidulans* strains generated in this article will be shared on reasonable request to the author.

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
