# Peer review of "Transcription Factors in the Fungus Aspergillus nidulans: Markers of Genetic Innovation, Network Rewiring and Conflict between Genomics and Transcriptomics"

_jof, 2021, doi:10.3390/jof7080600_

Round 1
Reviewer 1 Report
This manuscript expands, describes, and analyzes the list of transcription factors (the TFome) of the model fungus Aspergillus nidulans. Key findings include proposing that dispersed duplication is the likely mechanism for the expansion of Zn2Cys6 TF family members in A. nidulans, important annotation errors likely exist in the reference genome for a number of TFs, and a new regulator of growth and development is identified and characterized.
In general, this was a well-presented paper, and it will be informative to the Aspergillus and TF community. My major concerns are a lack of quantitative (and in some cases qualitative) criteria for a few analyses and a better rationale is needed for why the functional work in section 3.6 was included in this manuscript as it does not seem very well connected to the rest of the work.
Specifically:
Major Comments:
- L94-95: What criteria were used for adding a new TF to the list in Table S2?
- Can the author make a list of publications that provided new TFs as a new supplementary table or instead add a new column with this information to Table S2?
- Similarly, what analysis has FungiDB carried out or criteria have they used that identified TFs that were not originally found by Wortman, et al. 2009?
- Line 109: How were “outlier IPDs” defined?
- Figure 2A: What criteria were used to make these groups? No quantitative criteria are mentioned in the text. For example, something like “5 or more reads in two or more experiments are needed in order for a region to be considered exonic” (or something similar). Or for the opposite case, something like “fewer than 2 reads in the majority of experiments analyzed are needed in order to consider a region an intron” (or something similar).
- Lines 380-382: Were any quantitative criteria used for identifying these conditions?
- Lines 408-411: What was the criteria for focusing on the bHLH family?
- Section 3.6: I’m confused as to why this section and analyses was included in the paper. The rationale for it (lines 586-588: exploring a relationship between nitrogen starvation and the induction of development) does not seem to be related with the other TF-focused studies in this manuscript. More rationale or explanation is needed.
Minor Comments:
- In the Methods section and other applicable locations, the author should include versions of programs if possible, including for FungiDB.
- Line 191-192: Italicize Aspergillus
- Line 266: “…probably…”. Remove ambiguity. Either intron retention was or wasn’t the most common alternative splicing form.
- Figure 2D: The dotted orange squares are highlighting the ID events in the top panel but are highlighting an amplified region in the bottom two panels. For clarity, please change the color of the colored boxes highlighting the ID events in the top panel and add this description to the figure legend.
- Line 278-280: Clarify where the currently annotated start codon is located.
- Figure 2E: Similar to 2D, there are more dotted orange boxes than there are amplified regions in bottom panels. Please clarify the figure legend and/or the panel itself.
- Figure 2F: There is a dotted orange box present in the amplified box. Consider changing the color of this box and mentioning in the figure legend that this dotted box is highlighting the properly included intron.
- Is there a mutation or SNP in the intron that is found in the RNAseq data but not in the reference genome in this amplified region? If so, this should be mentioned in the figure legend and/or main text.
- Figure 2G: based on your nomenclature, should some of the “Sp” and “St” in this panel be changed to “pSp” or “aSP”? Delineating the difference between the two sets of St/Sp would be helpful.
- Figure 2I: There appears to be a stop codon two amino acids downstream of the new start codon in the middle reading frame shown in the panel. Please clarify the image or describe in the manuscript why this stop codon would not be used in the new annotation you are recommending.
- Have you submitted any of these new annotations to FungiDB or any other database that contains the nidulans reference genome and annotation?
- Lines 422-424: What part of Figure 5A shows the taxa of the sequence? Some method of showing which family/genus/species a TF was from would be useful. Maybe a track around the outside of the tree.
- Are these bHLH TFs conserved in other taxonomic families as well?
- Line 426-431: Were sequences for the new, properly annotated gene/protein used for these BLAST and HMMER searches?
- Lines 488-489: Why does this suggest they aren’t species-specific? Couldn’t these duplications have only occurred in nidulans?
- Lines 543-553: Why are the Msn homologs specifically mentioned here? Figure 7A appears to show many proteins as having just as similar sequences to BrlA in their DNA binding residues?
- Figure 7A – What does the red, dashed box signify?
- Figure S6C: Zn2Cys6 TFs appear to be the 3rd most-prevalent type of genes on this bar plot, not the first. Please clarify.
- Line 767: I think the period between FacB and PacC should be a comma.
Author Response
This manuscript expands, describes, and analyzes the list of transcription factors (the TFome) of the model fungus Aspergillus nidulans. Key findings include proposing that dispersed duplication is the likely mechanism for the expansion of Zn2Cys6 TF family members in A. nidulans, important annotation errors likely exist in the reference genome for a number of TFs, and a new regulator of growth and development is identified and characterized.
In general, this was a well-presented paper, and it will be informative to the Aspergillus and TF community. My major concerns are a lack of quantitative (and in some cases qualitative) criteria for a few analyses and a better rationale is needed for why the functional work in section 3.6 was included in this manuscript as it does not seem very well connected to the rest of the work.
-First of all, thanks to this reviewer for the comments and the contribution to the improvement of the work.
Specifically:
Major Comments:
- L94-95: What criteria were used for adding a new TF to the list in Table S2?
- Can the author make a list of publications that provided new TFs as a new supplementary table or instead add a new column with this information to Table S2?
- Similarly, what analysis has FungiDB carried out or criteria have they used that identified TFs that were not originally found by Wortman, et al. 2009?
-This piece of text has been rewritten (Lines 93-103 and lines 216-217 in the updated version). Fundamentally, TFs that were not originally included in the Table by Wortman and colleagues (2009) were found using the “protein features and properties: InterPro domain” function in the FungiDB database. The lists provided by the FungiDB database were compared to that published by Wortman et al. The sequence and predicted functional domains of those potential TFs not included in the table by Wortman et al were checked again in the FungiDB database and with the bioinformatics tools described in the next section (2.2), in order to include them or not in the present work.
- Line 109: How were “outlier IPDs” defined?
-The rough tree generated by MAFFT (Figure S1 in the first version) has been deleted. I simply intended to show a tree-like figure with all the potential IPD domains identified but that tree-like representation could misguide the reader
- Figure 2A: What criteria were used to make these groups? No quantitative criteria are mentioned in the text. For example, something like “5 or more reads in two or more experiments are needed in order for a region to be considered exonic” (or something similar). Or for the opposite case, something like “fewer than 2 reads in the majority of experiments analyzed are needed in order to consider a region an intron” (or something similar).
-In my opinion, the variety of cases that can be found when analyzing gene annotations is much higher than the single criteria suggested by this reviewer. The annotations of the FungiDB have been reviewed one by one and manually and there are cases in which with very few reads the position of the introns can be confirmed and other cases in which despite having multiple reads, it is more difficult to predict if a specific intron is processed as annotated or not. This is why, in all this cases, Tables S2 and S5 suggest a revision of the annotation of specific genes but usually cannot give a clear answers. That is why in the text corresponding to Figure 2A, only approximate figures are given for correctly or incorrectly annotated genes, or those that cannot be assessed based on RNA-seq reads. But the general conclusion is clear in my opinion: there is a significant subset of annotations that should be reviewed, independently of the exact number associated with each group. I have reviewed Table S2 and Table S5 for the groups of probably incorrectly annotated genes as well as those that cannot be assessed. Figure 2A has been updated accordingly.
- Lines 380-382: Were any quantitative criteria used for identifying these conditions?
-It was a qualitative assumption. This is why the word “suggest” was included in the sentence: “Figure 3 and Figure S2 also suggest that specific culture conditions and genetic backgrounds…”. The sentence has not been modified.
- Lines 408-411: What was the criteria for focusing on the bHLH family?
-In the case of bHLH-type TFs, for example AN0396, it was observed that the number of orthologs was limited and included another bHLH-type TF of A. nidulans, AN5078. This suggested the possibility of these two bHLH-type TFs being duplications and decided to carry out a deeper analysis of this family of TFs. The text has been rewritten (lines 428-430).
- Section 3.6: I’m confused as to why this section and analyses was included in the paper. The rationale for it (lines 586-588: exploring a relationship between nitrogen starvation and the induction of development) does not seem to be related with the other TF-focused studies in this manuscript. More rationale or explanation is needed.
-The present manuscript and a previous review of my group (Etxebeste et al., 2019) analyzed how new TFs emerge, how this can influence specific cellular processes and GRNs, or connections among GRNs. In section 3.6 of this manuscript, I just have tried to redirect and extend these evolutionary perspective and basic bioinformatics approach towards the identification of TFs playing key roles in growth in development. The hypothesis is that if specific GRNs are connected (metabolism, growth and development), transcriptomic analyses of a specific cellular process could help in the identification of regulators of the other network. Following the recommendation of this reviewer, I have tried to give a more detailed justification of the convenience of maintaining section 3.6 in this manuscript (lines 606-609).
Minor Comments:
- In the Methods section and other applicable locations, the author should include versions of programs if possible, including for FungiDB.
-Done. Check sections 2.1, 2.2 and 2.3.
- Line 191-192: Italicize Aspergillus
-Done (lines 206-207)
- Line 266: “…probably…”. Remove ambiguity. Either intron retention was or wasn’t the most common alternative splicing form.
-Modified to (line 281): Intron retention (IR) was a commonly found alternative splicing form.
- Figure 2D: The dotted orange squares are highlighting the ID events in the top panel but are highlighting an amplified region in the bottom two panels. For clarity, please change the color of the colored boxes highlighting the ID events in the top panel and add this description to the figure legend.
-Modified (lines 335 and 340).
- Line 278-280: Clarify where the currently annotated start codon is located.
-The region corresponding to the first annotated exon and the currently annotated start codon of AN11098 is not shown in the amplified region of panel 2E. To do that, I would have to include the predicted first intron in this amplification, which actually corresponds to the promoter of the gene. That would increase the size of the panel without adding any relevant information to the figure or to the main text.
- Figure 2E: Similar to 2D, there are more dotted orange boxes than there are amplified regions in bottom panels. Please clarify the figure legend and/or the panel itself.
-Done (lines 337 and 342).
- Figure 2F: There is a dotted orange box present in the amplified box. Consider changing the color of this box and mentioning in the figure legend that this dotted box is highlighting the properly included intron.
-Done. See the legend of Figure 2.
- Is there a mutation or SNP in the intron that is found in the RNAseq data but not in the reference genome in this amplified region? If so, this should be mentioned in the figure legend and/or main text.
-I do not see any SNP or mutation in the sequence of AN2782
- Figure 2G: based on your nomenclature, should some of the “Sp” and “St” in this panel be changed to “pSp” or “aSP”? Delineating the difference between the two sets of St/Sp would be helpful.
-Modified. The codon annotated as the start codon has been highlighted as pSt (predicted start codon; line 353)
- Figure 2I: There appears to be a stop codon two amino acids downstream of the new start codon in the middle reading frame shown in the panel. Please clarify the image or describe in the manuscript why this stop codon would not be used in the new annotation you are recommending.
-The stop codon indicated by this reviewer is located within the intron, which according to RNA-seq data, is processed. Processing of this intron is clear and it is not necessary to mention the presence of this stop codon.
- Have you submitted any of these new annotations to FungiDB or any other database that contains the nidulans reference genome and annotation?
-No, I have not. The aim of this section in the present manuscript is to highlight that there is an important proportion of annotations of genes encoding TFs that should be updated (and this statement can be extended to the whole set of genes of A. nidulans; not shown), but not to re-annotate each and every of them in detail. The author believes that this work corresponds to those researchers intending to functionally characterize those TFs in the future, and Table S2 in this manuscript gives valuable information to guide these re-annotations (together with the set of tools available in the FungiDB database).
- Lines 422-424: What part of Figure 5A shows the taxa of the sequence? Some method of showing which family/genus/species a TF was from would be useful. Maybe a track around the outside of the tree.
-The species the proteins included in Figure 5A correspond to are described in the main text. The accession number of the sequences in the tree are preceded by a code that indicates the species each sequence belongs to. Asydo, for example, indicates A. sydowii, and Anidu, A. nidulans. It has been described in the legend of the figure (lines 459-460).
- Are these bHLH TFs conserved in other taxonomic families as well?
-Of course, the conservation pattern of A. nidulans bHLH-type TFs is variable, and there are some of them showing conservation in more distant clades (for example, AN7661). However, the aim of the present work was to track the emergence of those with a limited phylogenetic distribution (AN2392, AN0396 and AN5078). That is why I focused the analysis on the family Trichocomaceae.
- Line 426-431: Were sequences for the new, properly annotated gene/protein used for these BLAST and HMMER searches?
-Yes, it was. See line 446 in the updated version of the manuscript.
- Lines 488-489: Why does this suggest they aren’t species-specific? Couldn’t these duplications have only occurred in nidulans?
-The reviewer is right and the second part of the sentence (“…which contrasts with the aforementioned general idea that these clusters were observed to be highly specific.”) has been deleted.
- Lines 543-553: Why are the Msn homologs specifically mentioned here? Figure 7A appears to show many proteins as having just as similar sequences to BrlA in their DNA binding residues?
-The reviewer is right when states that besides Msn orthologs there are many proteins with high conservation of the DNA-binding residues. However, the reference for which the amino acids binding the DNA target sequences (5´-A/CGGGG-3´) have been experimentally determined are Msn orthologs of S. cerevisiae. This is why Msn orthologs need to be specifically mentioned here.
- Figure 7A – What does the red, dashed box signify?
-The red, dashed square has been deleted.
- Figure S6C: Zn2Cys6 TFs appear to be the 3rd most-prevalent type of genes on this bar plot, not the first. Please clarify.
-The top Interpro domains are ordered in Figure S6C based on the enrichment of this domain (de-regulated in this comparison) compared to the total number proteins with this domain in A. nidulans. However, the graph indicates the number of genes encoding these InterPro domains in the list. It has been described in the legend of the figure.
- Line 767: I think the period between FacB and PacC should be a comma.
-The reviewer is right. Corrected (line 788).
Reviewer 2 Report
Congratulations to the esceptionally good research concept and results of the MS.
Please find my few comments on your MS:
The usage of bioinformatic method seems to be correct, but more comments about their use may be suggestible.
Abbreviations as a new heading between the Abstract and Introduction is recommended, because many of abbreviations are presented. Probably there is no comment to each of them, their comments are scatted, or not found at their first usage (as in the case of FPKM, TPM).
Is there need for small and pale cladograms on right sides of fig. 4. A & B? If yes, please comment both of them.
Ad 63. The usage of the taxon names Basidiomycota and Ascomycota may be more recommended.
Ad 404. Krizsán et al. is the correct citation.
Ad 535-536. What bootstep values are indicated by light purple circles? (Figure 6.) Please redefine it.
Ad 593-594. ni-trogen?
Ad 935. Names of Hungarian authors with not correct characters (cf with same names in 929)
Ad 1069-1070. Citation seems to be not completed.
Citations embeded in the text: 137-138, 384,
Please use italic for scientific names: 191, 987, 1000, 1065, 1107, 1145, 1146, 1149, 1154, 1164
Small typographical or grammatical errors: 960, 1068,
Author Response
Comments and Suggestions for Authors
Congratulations to the esceptionally good research concept and results of the MS.
Please find my few comments on your MS:
-Thanks to this reviewer for the comments.
The usage of bioinformatic method seems to be correct, but more comments about their use may be suggestible.
-I have checked sections 2.1, 2.2 and 2.3, and added more detailed information on the bioinformatics procedures followed.
Abbreviations as a new heading between the Abstract and Introduction is recommended, because many of abbreviations are presented. Probably there is no comment to each of them, their comments are scatted, or not found at their first usage (as in the case of FPKM, TPM).
-The manuscript has been reviewed and the meaning of all abbreviations added at their first usage.
Is there need for small and pale cladograms on right sides of fig. 4. A & B? If yes, please comment both of them.
-The cladograms have been deleted. The reviewer is right and there is no need for them.
Ad 63. The usage of the taxon names Basidiomycota and Ascomycota may be more recommended.
-Done (line 63).
Ad 404. Krizsán et al. is the correct citation.
-Corrected (line 421). Sorry for the mistake.
Ad 535-536. What bootstep values are indicated by light purple circles? (Figure 6.) Please redefine it.
-The MegaX software was used to generate phylogenetic trees. According to the website https://www.megasoftware.net/web_help_10/Bootstrap_Test_of_Phylogeny.htm, Mega uses Felsenstein's (1985) bootstrap test, which is evaluated using Efron's (1982) bootstrap resampling technique. I have included a reference in Materials and Methods, as this reviewer recommended a more detailed explanation on the usage of bioinformatics methods.
Ad 593-594. ni-trogen?
-Apparently, it is the format of the text in Journal of Fungi, and there are multiple cases throughout the manuscript.
Ad 935. Names of Hungarian authors with not correct characters (cf with same names in 929)
-Corrected (lines 952 and 958). Again, sorry for the mistake.
Ad 1069-1070. Citation seems to be not completed.
-Corrected.
Citations embeded in the text: 137-138, 384,
-Corrected.
Please use italic for scientific names: 191, 987, 1000, 1065, 1107, 1145, 1146, 1149, 1154, 1164
-Corrected.
Small typographical or grammatical errors: 960, 1068,
-Corrected.